# OUTLIER PRESERVING DISTRIBUTION MAPPING AUTOENCODERS

## ABSTRACT

State-of-the-art deep outlier detection methods map data into a latent space with the aim of having outliers far away from inliers in this space. Unfortunately, this is shown to often fail the divergence penalty they adopt pushes outliers into the same high-probability regions as inliers. We propose a novel method, OP-DMA, that successfully addresses the above problem. OP-DMA succeeds in mapping outliers to low probability regions in the latent space by leveraging a novel Prior-Weighted Loss (PWL) that utilizes the insight that outliers are likely to have a higher reconstruction error than inliers. Building on this insight, explicitly encourages outliers to be mapped to low-propbability regions of its latent by weighing the reconstruction error of individual points by a multivariate Gaussian probability density function evaluated at each point's latent representation. We formally prove that OP-DMA succeeds to map outliers to low-probability regions. Our experimental study demonstrates that OP-DMA consistently outperforms state-of-art methods on a rich variety of outlier detection benchmark datasets.

## 1 INTRODUCTION

**Background.** Outlier detection, the task of discovering abnormal instances in a dataset, is critical for applications from fraud detection, error measurement identification to system fault detection (Singh & Upadhyaya, 2012). Given outliers are by definition rare, it is often infeasible to get enough labeled outlier examples that are represetnative of all the forms the outliers could take. Consequently, unsupervised outlier detection methods that do not require prior labeling of inliers or outliers are frequently adopted (Chandola et al., 2009).

**State-of-Art Deep Learning Methods for Outlier Detection.** Deep learning methods for outlier detection commonly utilize the reconstruction error of an autoencoder model as an outlier score for outlier detection (Sakurada & Yairi, 2014; Vu et al., 2019). However, directly using the reconstruction error as the outlier score has a major flaw. As the learning process converges, both outliers and inliers tend to converge to the average reconstruction error (to the same outlier score) – making them indistinguishable (Beggel et al., 2019). This is demonstrated in Figure 1a, which shows that the ratio of average reconstruction error for outliers converges to that of the inliers.

To overcome this shortcoming, recent work (Beggel et al., 2019; Perera et al., 2019) utilizes the distribution-mapping capabilities of generative models that encourage data to follow a prior distribution in the latent space. These cutting-edge methods assume that while the mapping of inlier points will follow the target prior distribution, outliers will not due to their anomalous nature. Instead, outliers will be mapped to low-probability regions of the prior distribution, making it easy to detect them as outliers (Beggel et al., 2019; Perera et al., 2019).

However, this widely held assumption has been shown to not hold in practice (Perera et al., 2019). Unfortunately, as shown in Figure 1b, both inliers and outliers are still mapped to the same high probability regions of the target prior distribution, making them difficult to distinguish.

**Problem Definition.** Given a given dataset $X \in \mathbb{R}^M$ of multivariate observations, let $f : \mathbb{R}^M \to \mathbb{R}^N$, $N \leq M$, be a function from the multivariate feature space of $X$ to a latent space $f(x) \in \mathbb{R}^N$ such that $f(X) \sim P_Z$, where $P_Z$ is a known and tractable prior probability density function. The dataset $X \in \mathbb{R}^M$ is composed as $X = X_O + X_I$, where $X_O$ and $X_I$ are a set of outlier and inlier points, respectively. During training, it is unknown whether any given point $x \in X$ is an outlier

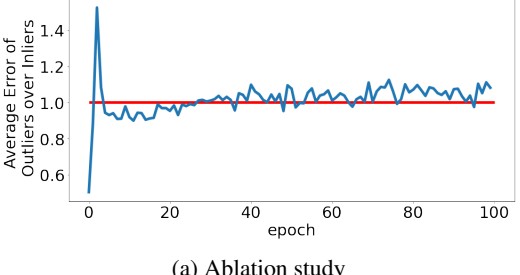

(a) Ablation study

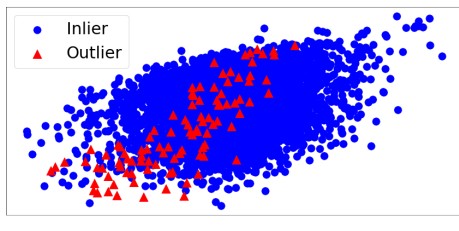

(b) Inliers and outliers in latent space of leading distribution-mappign method

Figure 1: Data Set used in (a) and (b) is Inliers taken from MNIST "1"s while outliers are MNIST "0"s, such that the outliers account for roughly 20% of the total data. (a) Left plot shows average reconstruction error of outliers over average reconstruction error of inliers during the training of a standard autoencoder. of the total data. As the plot shows, the ratio of errors for outliers to inliers goes to 1, meaning outliers are difficult to distinguish from inliers after training. (b) The right plot shows inliers and outliers in the 2-dimensional latent space of a Wasserstein Autoencoder (a popular type of distribution mapping autoencoder). As seen, the outliers are in high-probability regions of the latent space and are thus difficult to separate from the inliers.

or an inlier. Intuitively, our goal is to find a function $f$ that maps instances of a dataset $X$ into a latent space $S$ with a known distribution, such that outliers are mapped to low probability regions and inliers to high probability regions. More formally, we define *unsupervised distribution-mapping outlier detection* as the problem of finding a function $f^*$ with the aforementioned properties of $f$ such that we maximize the number of outliers $x_o \in X_O$ and inliers $x_i \in X_I$ for which $P_Z(f^*(x_o)) < P_Z(f^*(x_i))$ holds.

**Challenges.** To address the open problem defined above, the following challenges exist:

1. **Overpowering divergence penalty.** Intuitively, distribution mapping methods utilize a divergence penalty to achieve a latent space mapping of input data that has a high probability of following a target prior distribution. While the data overall should follow this prior distribution, a solution must be found to instead maps outliers to low-probability regions of the prior. Having the data match the prior overall, while having outliers mapped to low probability regions of the prior creates a conflict, as the two tasks are diametrically opposed. To achieve such a mapping requires overpowering the divergence penalty in order to map outliers to low probability regions in the latent space.

2. **Unknown outlier status.** In unsupervised outlier detection, during training points do not have any labels indicating whether they are outliers or inliers. This unsupervised scenario, while common in practice (Singh & Upadhyaya, 2012), makes it challenging to design strategies that explicitly coerce outliers to be mapped to low-probability regions.

**Our OP-DMA Approach.** In this work, we propose the Outlier Preserving Distribution Mapping Autoencoder (OP-DMA). Our core idea is to propose a novel Prior Weighted Loss (PWL) function that solves the two conflicting tasks of mapping the input data to a prior distribution while encouraging outliers to be mapped to low probability regions of that prior. This PWL directly addresses the shortcomings of the existing distribution mapping outlier detection methods (Vu et al., 2019; Perera et al., 2019), and to the best of our knowledge is the first unsupervised cost function that explicitly encourages outliers to be mapped to low probability regions.

We assume that outliers will have a high reconstruction error during the initial stages of training, which causes the PWL to place them in low-probability (low PDF) regions in the latent space. This way, PWL overcomes the challenge of *overpowering the divergence penalty*. It succeeds in mapping outliers to low-probability regions (far from the mean of the latent distribution) even though each input point's *outlier status is unknown*. Our OP-DMA framework is pluggable, meaning off-the-shelf distance-based outlier methods can be flexibly plugged in post-transformation.

**Our key contributions** are as follows:

1. Propose OP-DMA, a novel distribution-mapping autoencoder that effectively separates outliers from inliers in the latent space without knowing nor making assumptions on the original distribution of the data in the feature space.

2. Design the Prior-Weighted Loss (PWL), which when coupled with a divergence penalty encourages outliers to be mapped to low-probability regions while inliers are mapped to high-probability regions of the latent space of an autoencoder.

3. Provide rigorous theoretical proof that the optimal solution for OP-DMA places outliers further than inliers from the mean of the distribution of the data in the latent space.

4. Demonstrate experimentally that OP-DMA consistently outperforms other state-of-art outlier detection methods on a rich variety of real-world benchmark outlier datasets.

**Significance:** OP-DMA is a versatile outlier detection strategy as it can handle input data that has arbitrary distributions in the feature space, while not making any distance or density assumptions on the data. To the best of our knowledge, we are the first to propose a loss function that explicitly encourages outliers to be mapped to low-probability regions while inliers are mapped to high probability regions. Our PWL approach is pluggable, and can easily be incorporated into alternate outlier detectors. Our ideas could also spur further research into various prior weighted loss functions.

## 2 Related Work

State-of-the-art deep outlier detection methods fall into one of three categories: *1) Autoencoders coupled with classic outlier detectors* (Erfani et al., 2016; Chalapathy et al., 2018), *2) Reconstruction error-based outlier detection methods* (Zhou & Paffenroth, 2017; Chen et al., 2017; Sabokrou et al., 2018; Xia et al., 2015), or *3) Generative outlier detection methods* (Perera et al., 2019; Vu et al., 2019; Liu et al., 2019).

*1) Autoencoders coupled with classic outlier detectors* project data into a lower dimensional latent space before performing outlier detection on that latent representation. These methods make the strict assumption that outliers in the original space will remain outliers in the latent space. Further, they fail to explicitly encourage this in the mapping function.

*2) Reconstruction error-based outlier detection methods* utilize the reconstruction error of an autoencoder network to identify outliers. They typically use the reconstruction error directly as the anomaly score (An & Cho, 2015). In more recent work, they try to separate outliers into a separate low-rank matrix analogous to RPCA (Zhou & Paffenroth, 2017) or they introduce a separate discriminator network (Sabokrou et al., 2018). However, as shown in (Beggel et al., 2019), for autoencoders the reconstruction error of outliers often converges to that of inliers. This negatively impacts the performance of such reconstruction error methods.

*3) Generative outlier detection methods* leverage deep generative models (Goodfellow et al., 2014; Kingma & Welling, 2013) to generate the latent space such that the distribution of the latent space is encouraged to match a known prior so that thereafter an appropriate outlier method for the prior can be applied (Vu et al., 2019) to the latent space, or a discriminator can identify outliers in the latent space (Vu et al., 2019) or both the latent space and reconstructed space (Perera et al., 2019). However, as discussed in Section 1, in practice both inliers and outliers are both mapped to the prior distribution as outliers that are mapped to low-probability regions will generally incur a high cost from the divergence term which matches the latent distribution to the prior.

OP-DMA shares characteristics with each of these three categories. However, unlike the other methods in these categories, OP-DMA actively *encourages* outlier to be mapped to low-probability regions instead of just assuming that this will be the case. OP-DMA is is a *generative outlier method* that uses the *reconstruction error* to encourage outliers to be mapped to low-probability regions. Further, it can flexibly be paired with nearly any *classic outlier detector* after distribution mapping.

## 3 Proposed Approach: OP-DMA

**Overview of approach.** OP-DMA consists of three main components:

1. *A distribution mapping autoencoder (DMA)* that OP-DMA utilizes to map a dataset $X$ from the feature space $\mathbb{R}^M$ into a lower dimensional latent space $\mathbb{R}^N$, such that the distribution of

the encoded data in the lower dimensional latent space has a known probability distribution $P_Z$. This step is crucial as it makes it easy for OP-DMA to easily identify low probability regions of the latent space (outliers should be mapped here). This can be done because after the distribution mapping, we can explicitly calculate the Probability Density Function (PDF) of the latent space so long as we selected a prior distribution with a known PDF.

2. *A novel Probability-Weighted Loss (PWL) function* for distribution mapping that encourages outliers to be mapped to low-probability regions of the latent space, solving both the challenges of **overpowering divergence penalty** and **unknown outlier status**.

3. *An traditional outlier detection method* is used to identify outliers in the transformed latent space. The choice of outlier detection method is flexible as long as it is amenable to the prior distribution $P_Z$ selected in step 1 of OP-DMA. For instance, when a Gaussian distribution is used for the prior, then OP-DMA utilizes a classical distance-based outlier detection method for step 3. These steps are described in the following subsections and illustrated in Figure 2.

### 3.1 DISTRIBUTION MAPPING AUTOENCODER (DMA)

In order to use prior-weighting to map outliers to low-probability regions of a known PDF in a latent space, our distribution mapping method must meet *two design requirements*:

1. A one-to-one mapping between each original data point, its latent representation and the reconstructed data point must be established so that each data point's reconstructed data point is unique and can be determined, and vice versa.

2. The divergence term must impose a cost based on how well a batch of latent data points match the prior overall, rather than requiring individual data points to have a high probability of being a draw from the prior.

To meet these requirements, we select the *Wasserstein AutoEncoder (WAE)* (Tolstikhin et al., 2017) as the foundation for our distribution mapping. WAEs are distribution-mapping autoencoders that minimize the *Wasserstein distance* between the original data and its reconstruction, while mapping the input data to a latent space with a known prior distribution. To see why we base our distribution-mapping technique on this method, consider the WAE objective function for encoder network $Q$ and decoder network $G$:

$$W_c^\lambda(X,Y) = \overbrace{\inf_Q \mathbb{E}_{P_X} \mathbb{E}_{Q(Z|X)}[c(X,G(Z))]}^{\text{Reconstruction Error}} + \overbrace{\lambda \mathcal{D}(P_Q, P_Z)}^{\text{Divergence Penalty}}. \quad (1)$$

The first term on the right hand side of Equation 1 corresponds to the reconstruction error between the input data and reconstructed data for cost function $c$. The second term $\mathcal{D}$ is a divergence penalty between the distribution of the latent space and the prior distribution, with $\lambda$ a constant weight term that determines how much that divergence is penalized. Let us deterministically produce the latent representation $Q(X)$ and output $G(Q(X)|X)$ (by using $Q(X) = \delta_{\mu(X)}$, where $\mu$ is some function

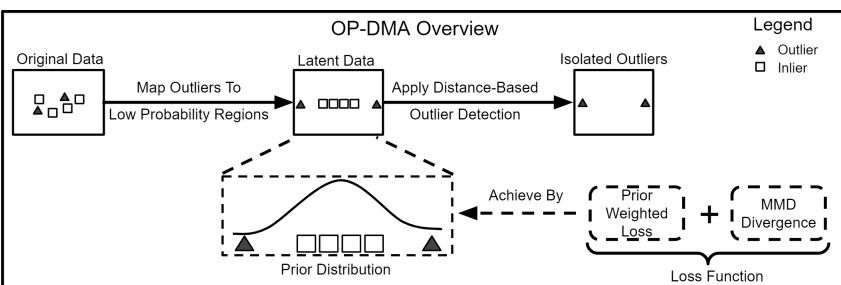

Figure 2: An overview of OP-DMA. Data is mapped to latent space with prior distribution using a Prior-Weighted Loss (PWL), which encourages outliers to be mapped to low-probability regions. This allows for distance-based outlier detection.

mapping input data set $X$ to $Q(X)$, for instance). It is now clear why Wasserstein autoencoders are an appropriate choice to model our distribution mapping method, as the reconstruction error term $\mathbb{E}_{P_X}\mathbb{E}_{Q(Z|X)}[c(X, G(Z))]$ in Equation 1 represents a one-to-one correspondence between input data, its latent representation and the reconstructed output (meeting requirement 1). Additionally, $\mathcal{D}$ is a *batch-level cost term* that would be incurred if the latent representation of a batch doesn't match the prior distribution but doesn't require individual points to be mapped to a high probability region of the prior (meeting requirement 2). However, we note that WAEs unfortunately *do not* encourage outliers in the feature space to remain outliers in the latent space. Consider $\mathcal{D}$ to be a discriminator network. Then $\mathcal{D}$ is likely to learn a boundary around the high probability region of the prior distribution. Thus the encoder network Q will be penalized for mapping an outlier to a low probability region outside of the boundary found by D as the discriminator D would correctly identify it as a generated point.

## 3.2 Prior-Weighted Loss (PWL): Novel Loss Function for Outlier Coercion

We now describe our novel Prior-Weighted Loss (PWL) that tackles the above challenge of WAEs mapping outliers to high probability regions. The key idea is that outliers will *initially* have higher reconstruction error than inliers during training. This core idea draws from the area of anomaly detection using reconstruction probability (An & Cho, 2015). We thus propose the *Prior Weighted Loss (PWL)*, a novel cost term that weights each data point's reconstruction error term in Equation 1 by the point's *latent likelihood*, $P_Z(Q(x))$. The latent likelihood is the PDF of the latent space's prior distribution evaluated at its corresponding latent representation.

The prior weighted loss $c'$ is defined as $c' := c(x, G(Q(x))) \cdot P_Z(Q(X))$

As the latent likelihood is large in high probability regions and small in low probability regions by definition, points with a high reconstruction error that are mapped to high-probability regions will be penalized more than those with high reconstruction error that are mapped to low probability regions. Since outliers are assumed to result in a high reconstruction error (at least during early training epochs), by reducing the penalty to the network for poorly reconstructed points that have been mapped to low-probability regions of the prior, the network is encouraged to map outliers to these low-probability regions. We now introduce our OP-DMA objective $W_{c'}^\lambda$ as:

$$W_{c'}^\lambda = \overbrace{\inf_{Q:P_Q=P_Z} \mathbb{E}_{P_X}\mathbb{E}_{Q(Z|X)}[c'(X, G(Z))]}^{\text{Prior weighted loss}} + \overbrace{\lambda\mathcal{D}(P_Q, P_Z)}^{\text{Divergence penalty}} \tag{2}$$

Since we have significantly modified the reconstruction error term in the Wasserstein autoencoder loss function, a natural question is whether or not OP-DMA still corresponds to an autoencoder. Specifically, will the decoder's output still match the input data to the encoder? If this does not hold, two issues could arise: 1) The latent features learned by the network might be unrelated to the input, and hence useless in cases where it is desirable to use the latent representation in a downstream task. 2) More importantly for our outlier detection task, if the network is no longer encouraged to reconstruct the input, the crucial property that outliers will have a higher reconstruction error may no longer hold. In such a case, the "reconstruction error" may be meaningless. Fortunately, we can show that our OP-DMA loss function still corresponds to a Wasserstein divergence between the input and reconstructed distributions (Theorem 1). For this, we must demonstrate that is that the prior-weighted cost $c'$ meets the requirements of a Wasserstein divergence's cost function, namely, that $c'(x_1, x_2) \geq 0$ $(\forall\, x_1, x_2 \in supp(P))$, $(c'(x, x) = 0)$ $(\forall\, x \in supp(P))$, and $\mathbb{E}_\gamma[c'(x_1, x_2)] \geq 0$ $(\forall\, \gamma \in \Gamma[P, P_Z])$

**Theorem 1.** *Let $W_c$ be a Wasserstein distance. Then $W_{c'}$ is a Wasserstein distance, with c' the prior-weighted c.*

## 3.3 Unsupervised Statistical Outlier Detection Method

Intuitively, an ideal mapping would place all inliers within regions where the latent likelihood is greater than some value $V$, and all outliers into some alternate regions where the latent likelihood is less than that value $V$. The core result fundamental to our work is thus that this scenario is indeed the optimal solution for the loss function of OP-DMA as stated in Theorem 2.

| Dataset | # Features | # Datapoints | % Outliers | | Dataset | # Features | # Datapoints | % Outliers |
|---|---|---|---|---|---|---|---|---|
| Satellite | 36 | 6435 | 32% | | Lympho | 18 | 148 | 4.1% |
| Pima | 8 | 768 | 35% | | Musk | 166 | 3062 | 3.2% |
| WBC | 30 | 278 | 5.6% | | Thyroid | 6 | 3772 | 2.5% |
| Arrythmia | 274 | 452 | 15% | | Satimage-2 | 36 | 5803 | 1.2% |
| Breastw | 9 | 638 | 35% | | Cover | 10 | 286048 | 0.9% |
| Letter | 32 | 1600 | 6.25% | | Fever | 36 | 5293 | 0.2% |
| Cardio | 21 | 1831 | 9.6% | | MNIST | 784 | 6365 | 1% |

Table 1: Description of real-world datasets' dimensionality, size, and outlier percentage. Most datasets taken from the standard ODDs database[1], while RC Flu was taken from the Reality-Commons Social Evolution database[2]. We also evaluate on the well-known MNIST[3] dataset.

**Theorem 2.** *Let $Q$ be an encoder network such that $\mathcal{D}(P_Q, P_Z, \mathcal{F}) = 0$, where $\mathcal{D}(A, B, \mathcal{F})$ is the Maximum Mean Discrepancy between A and B, $\mathcal{F}$ is the set of bounded continuous functions and $P_Z = \mathcal{N}(\mathbf{0}, \mathbf{\Sigma})$. Let us consider or dataset $X$ as a centered random variable, $X : \Omega \to \mathbb{R}^n$, $X \sim P_X$. Let $X(A)$, $A \subset \Omega$, be outliers and let $H = \Omega - A$ be the inliers, where $\int_{X(A)} p_X(x)dx = \alpha$. Further, let $c'(a, G(Q(a)) > c'(h, G(Q(h)) \,\forall\, a \in X(A), h \in X(H)$. Then, the optimal solution of OP-DMA is to map such that $\|Q(X(A))\|_{mahalanobis} \geq \delta$ and $\|Q(X(H))\|_{mahalanobis} < \delta$, where*

$$\delta = \sqrt{\int_0^{1-\alpha} \frac{t^{-n/2-1}e^{\frac{1}{2t}}}{2^{\frac{n}{2}}\Gamma(\frac{n}{2})}dt} \tag{3}$$

This important result implies that after transformation with OP-DMA outliers can be separated from inliers using a simple distance metric. This lays a solid foundation for a simple yet effective outlier detection scheme. Namely, we first transform the dataset $X$ to a latent representation with a multivariate Gaussian prior distribution, as justified by Theorem 2. Then, as Equation 3 states, outliers can be isolated using a simple distance-based approach. More specifically, any standard outlier detection method that finds outliers in Gaussian distributions (e.g. EllipticEnvelope method (Rousseeuw & Driessen, 1999)) can be used to find outliers in the latent space.

### 3.4 PULLING IT ALL TOGETHER: UNSUPERVISED OUTLIER DETECTION USING OP-DMA

OP-DMA, our end-to-end outlier detection approach, is now summarized. First, the input data is transformed to match a prior distribution with a distribution mapping autoencoder using our novel Prior-Weighted Loss (PWL) (Equation 2). We chose this prior to be a multivariate Gaussian distribution with 0 mean and identity covariance, as justified by Theorem 2. Then, an Elliptic Envelope (Rousseeuw & Driessen, 1999) is used to identify outliers. The outlier detection process is outlined in Appendix A.3. We use the unbiased estimator of Maximum Mean Discrepency (MMD) from (Gretton et al., 2012) for the divergence term. For the kernel $k$ of MMD, we use the inverse multiquadratics kernel as in (Tolstikhin et al., 2017) and *Mean Squared Error* (MSE) for $c$.

## 4 EXPERIMENTAL EVALUATION

**Compared Methods.** We compare OP-DMA to state-of-the-art distribution mapping outlier detection methods. These include methods that perform outlier detection on the latent space of a WAE (Tolstikhin et al., 2017), a VAE (Kingma & Welling, 2013), and an *Adversarial Autoencoder* (AAE) (Makhzani et al., 2015) – all with a Gaussian prior but they do not integrate our PWL idea. We test against MO-GAAL (Liu et al., 2019) and ALOCC (Sabokrou et al., 2018), two state-of-the-art deep generative outlier detection models. We also test against LOF (Breunig et al., 2000) and OC-SVM (Schölkopf et al., 2001), two popular state-of-the-art non-deep outlier detection methods.

**Data Sets.** We evaluated on a rich variety of real-world data sets from the ODDs [1] benchmark data store (Rayana, 2016). These datasets cover a wide range of dimensionality in the feature space from 6 to 274, and also different outlier contamination percentages from 0.2% to 32%.

---

[1] http://odds.cs.stonybrook.edu/

| Datasets!Methods | OP-DMA | WAE | VAE | AAE | MO-GAAL | ALOCC | LOF | OC-SVM |
|---|---|---|---|---|---|---|---|---|
| `Satellite` | **0.735** ±0.012 | 0.554 ±0.009 | 0.310 ±0.007 | 0.480 ±0.008 | 0.481 ±0.001 | 0.706 ±0.008 | 0.413 | 0.417 |
| `Pima` | **0.625** ±0.018 | 0.520 ±0.020 | 0.23 ±0.019 | 0.497 ±0.007 | 0.518 ±0.002 | 0.517 ±0.010 | 0.456 | 0.440 |
| `WBC` | **0.590** ±0.011 | 0.448 ±0.013 | 0.268 ±0.011 | 0.19 ±0.018 | 0.468 ±0.060 | 0.529 ±0.007 | 0.480 | 0.199 |
| `Arrythmia` | 0.531 ±0.017 | **0.601** ±0.015 | 0.201 ±0.010 | 0.294 ±0.010 | 0.518 ±0.011 | 0.457 ±0.020 | 0.464 | 0.254 |
| `Breastw` | **0.951** ±0.014 | 0.950 ±0.011 | 0.368 ±0.009 | 0.479 ±0.007 | 0.944 ±0.014 | 0.863 ±0.062 | 0.292 | 0.824 |
| `Letter` | 0.182 ±0.001 | 0.091 ±0.003 | 0.048 ±0.005 | 0.10 ±0.002 | 0.159 ±0.003 | 0.165 ±0.001 | **0.488** | 0.208 |
| `Cardio` | **0.590** ±0.013 | 0.290 ±0.012 | 0.221 ±0.008 | 0.204 ±0.009 | 0.542 ±0.032 | 0.432 ±0.071 | 0.208 | 0.323 |
| `Lympho` | 0.585 ±0.012 | 0.443 ±0.008 | 0.341 ±0.011 | 0.310 ±0.018 | 0.719 ±0.119 | **0.958** ±0.083 | 0.833 | 0.150 |
| `Musk` | 0.32 ±0.007 | 0.330 ±0.010 | 0.243 ±0.009 | 0.228 ±0.025 | **0.394** ±0.090 | 0.199 ±0.160 | 0.069 | 0.101 |
| `Thyroid` | **0.29** ±0.019 | 0.173 ±0.021 | 0.130 ±0.019 | 0.170 ±0.023 | 0.225 ±0.027 | 0.084 ±0.026 | 0.200 | 0.090 |
| `Satimage-2` | **0.860** ±0.039 | 0.176 ±0.013 | 0.148 ±0.007 | 0.535 ±0.015 | 0.814 ±0.049 | 0.818 ±0.026 | 0.122 | 0.019 |
| `Cover` | **0.200** ±0.002 | 0.142 ±0.001 | 0.069 ±0.004 | 0.070 ±0.001 | 0.032 ±0.009 | 0.050 ±0.003 | 0.0227 | 0.1194 |
| `Fever` | **0.854** ±0.031 | 0.786 ±0.027 | 0.274 ±0.006 | 0.485 ±0.014 | 0.560 ±0.030 | 0.720 ±0.094 | 0.459 | 0.568 |
| `MNIST` | **0.684** ±0.020 | 0.532 ±0.042 | 0.614 ±0.036 | 0.599 ±0.013 | 0.469 ±0.141 | 0.571 ±0.026 | 0.491 | 0.610 |

Table 2: Weighted F1 scores with 95% confidence interval for OP-DMA vs state-of-the-art methods on benchmark outlier detection datasets. Best performing method in bold, second-best underlined. No confidence intervals on LOG and OC-SVM as they are deterministic.

Table 1 breaks down the statistics of each dataset. We evaluate all methods on their ability to detect subjects who have a fever from smartphone sensible data using the MIT Social Evolution dataset (Madan et al., 2011) (RC Fever) [2] to demonstrate OP-DMA's effectiveness for mobile healthcare. Finally, we also evalue on the MNIST dataset [3]. We used all MNIST images of "7"s as inliers, and randomly sampled "0"s as outliers such that "0"s account for $\sim 1\%$ of the data. Since outlier detection is unsupervised without any supervised training phase, we perform outlier detection in an unsupervised manner on the entire dataset instead of having to introduce train/test splits. In each dataset, all points are labeled as either inlier or outlier as ground truth. We emphasize that these ground truth labels are only used for evaluation but not for training all methods.

**Metrics.** Due to the large class imbalance inherent to outlier detection, we use the F1 score as our performance metric (Lazarevic-McManus et al., 2008) as commonly used to evaluate outlier detection methods (An & Cho, 2015; Zhou & Paffenroth, 2017; Zong et al., 2018).

**Parameter Configurations of Methods.** Encoders and decoders of all methods consist of 3-layer neural networks, where the decoder in each pair mirrors the structure of its encoder. The number of nodes in the hidden layer of each network is a hyperparameter from {5, 6, 9, 15, 18, 100}. The number of nodes in the latent layer varies from {2, 3, 6, 9, 15}. The regularization parameter $\lambda$ is chosen such that the reconstruction error is on the same order of magnitude as the MMD error for the first epoch. We use the standard parameters of MO-GAAL from the authors' code [4]. We also use the standard configuration of ALOCC from the authors code [5], except we add an additional dense layer at the beginning of each subnetwork. We do this as ALOCC assumes input to be images of a certain shape. The additional dense layer transforms the input data from its original dimensionality

---

[2]http://realitycommons.media.mit.edu/socialevolution4.html

[3]http://yann.lecun.com/exdb/mnist/

[4]https://github.com/leibinghe/GAAL-based-outlier-detection

[5]https://github.com/khalooei/ALOCC-CVPR2018

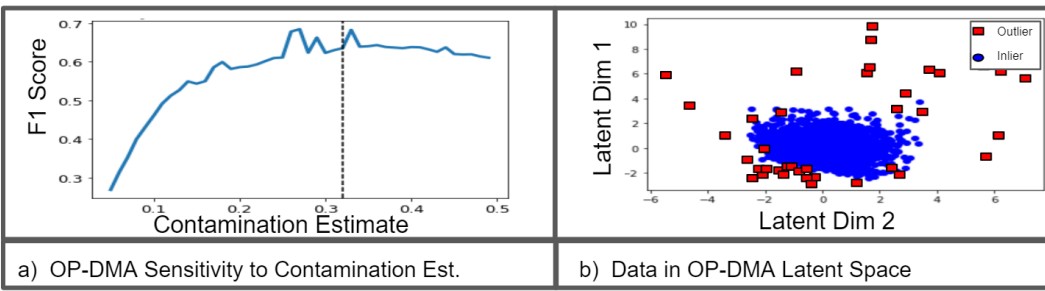

Figure 3: a) F1 score of OP-DMA for various values of the contamination parameter on the Satimage-2 dataset. b) Outliers and inliers in OP-DMA's latent space.

into the required shape. For these we use the standard parameters from Scikit-Learn.

**Experiment 1: Versatile Anomaly Detection.** We validate the versatility of our OP-DMA method by showing that our method consistently outperforms state-of-the-art methods on a rich variety of benchmark datasets. As shown in Table 2, OP-DMA outperforms the majority (9/13) of the other methods on the benchmark datasets. We see that OP-DMA's superior performance is not limited to datasets with either a high or low percentage of outliers. OP-DMA is the best performing method on the dataset with the largest ratio of outliers (Satellite) as well as that with the smallest ratio (Cover).

**Experiment 2: Sensitivity to Contamination Parameter.** The contamination parameter $\alpha$ is used to fit the standard outlier method, Elliptic Envelope, plugged into our OP-DMA framework on the encoded data after training. Thus, we test the sensitivity of EllipticEnvelope to the value of the contamination parameter by evaluating the F1 score of outlier detection on the Satellite dataset mapped by OP-DMA. The results (Figure 3 (a)) show that as long as this parameter is not significantly underestimated, the F1-score is robust to different values of the contamination parameter.

**Experiment 3: Verifying that Outliers are Mapped To Low-Probability Regions.** We transformed data from a multi-modal distribution in $\mathbb{R}^4$ consisting of a mixture of two Gaussians centered at (0,0,0,0) and (5,5,5,5) to a standard normal Gaussian in $\mathbb{R}^2$. Outliers in the original space were drawn from a uniform distribution and consisted of 2.4% of the total data. As Figure 3 b) shows, outliers are successfully mapped far from the inlier data points. Furthermore, the average value of the prior evaluated at the outlier points is 0.02, while the average for inliers is 0.08, confirming that outliers are mapped to lower-probability regions than inliers.

## 5 Conclusion

We have introduced OP-DMA, an autoencoder-based solution that unlike prior methods is truly outlier preserving in its distribution mapping method. That is, OP-DMA maps outliers in the feature space to low probability regions in the latent space in which a multivariate standard normal Gaussian prior distribution is enforced. Outliers are consequently easily identifiable in the latent space. Our experimental study comparing OP-DMA to state-of-the-art methods on a collection of benchmark outlier detection datasets shows that it consistently outperforms these methods on the majority of the datasets. We have also demonstrated that there is not a significant increase in running time between our method and state-of-the-art methods.

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

# A   APPENDIX

## A.1   PROOF OF THEOREM 1

*Proof.* Since $c$ is a Wasserstein divergence, we know that $c(x_1, x_2) \geq 0$ $(\forall\, x_1, x_2 \in supp(P))$, $(c(x, x) = 0)$ $(\forall\, x \in supp(P))$, and $\mathbb{E}_\gamma[c(x_1, x_2)] \geq 0$ $(\forall\, \gamma \in \Gamma[P, P_Z])$. Since $P_Z(z) \geq 0$ $(\forall\, z)$, $c'$ will also fulfill the three aforementioned properties of $c$. Thus, $W_{c'}$ is a Wasserstein divergence. $\square$

## A.2   PROOF OF THEOREM 2

*Proof.* The Mahalanobis distance of $Q(X)$ can itself be expressed as a random variable, $\delta = \sqrt{Q(X)\Sigma^{-1}Q(X)^T}$. Let $\Phi_\delta$ be the CDF of $\delta$. Then, $\Phi_\delta(1 - \alpha) = P(\delta \leq 1 - \alpha) = P(\delta^2 \leq (1 - \alpha)^2) = \Phi_{\delta^2}((1 - \alpha)^2)$.

Let $Y = Q(X)M^{-1}$, where $M^T M = \Sigma$ is the Choleski decomposition of the covariance $\Sigma$. Since $\mathcal{D}(P_Q, P_Z, \mathcal{F}) = 0$, and $\mathcal{D}(A, B, \mathcal{F}) = 0$ iff $A = B$, we thus know that $Q(X) \sim \mathcal{N}(\mathbf{0}, \boldsymbol{\Sigma})$. Thus since $Q(X)$ is normally distributed and centered, $Y$ is normally distributed with identity covariance. Since $\delta^2 = Q(X)\Sigma^{-1}Q(X)^T = YY^T$, $\Phi_{\delta^2}$ is the CDF of of the sum of squares of $n$ normally distributed variables with mean 0 and $\sigma = 1$. Thus, $\Phi_{\delta^2}$ is the Chi Squared distribution. The inverse Chi Squared CDF will thus give us the distance $\delta$ such that $1 - \alpha$ percent of the points are within $\delta = \sqrt{\int_0^{1-\alpha} \frac{t^{-n/2-1}e^{\frac{1}{2t}}}{2^{\frac{n}{2}}\Gamma(\frac{n}{2})}\,dt}$ Now, let us assume that for some parameter choice $\Theta'$ for $Q$ that $\alpha P(Q(X(A)|\Theta') \leq \delta) = \beta$, $\beta > 0$. Consequently, $(1 - \alpha)P(Q(X(H)|\Theta') > \delta) = \beta$, since $P(Q(X) > \delta) = \alpha$ and $\int_{X(A)} p_X(x)dx = \alpha$. Conversely, let us assume that there is a parameter configuration $\Theta$ such that $\alpha P(Q(X(A)|\Theta) \leq \delta) = 0$ and so $(1 - \alpha)P(Q(X(H)|\Theta) > \delta) = 0$.

Since $P_Z \sim \mathcal{N}(\mathbf{0}, \boldsymbol{\Sigma})$, $P_Z(d_1) < P_Z(d_2)$ for $\|d_1\|_{mahalanobis} > \|d_2\|_{mahalanobis}$. Thus, since we assume $c(a, G(Q(a)) > c(h, G(Q(h)) \,\forall\, a \in X(A), h \in X(H)$, then

$$\mathbb{E}_{P_X}\mathbb{E}_{Q(Z|X)}c'(x_p, G(Q(x_p|\Theta'))) = \mathbb{E}_{P_X}\mathbb{E}_{Q(Z|X)}c(x_p, G(Q(x_p|\Theta')))P_Z(x_p)$$
$$> \mathbb{E}_{P_X}\mathbb{E}_{Q(Z|X)}c(x_p, G(Q(x_p|\Theta)))P_Z(x_p) = \mathbb{E}_{P_X}\mathbb{E}_{Q(Z|X)}c'(x_p, G(Q(x_p|\Theta))).$$

Thus, the optimal solution for OP-DMA's cost function is one that maps outliers to regions with a larger Mahalanobis distance than that of inliers. $\square$

## A.3   OP-DMA ALGORITHM

---

**Algorithm 1:** Unsupervised Outlier Detection with OP-DMA

---

**Require:** Regularization coefficient $\lambda$
Contamination parameter $\alpha$
Initialized encoder network $Q_\Phi$ and decoder network $G_\Theta$ with random weights $\Phi$ and $\Theta$
Dataset $X$
**while** $\Theta$, $\Phi$ *not converged* **do**
    Sample $\{x_1, x_1, ..., x_n\}$ from $X$, $\{z_1, z_1, ..., z_n\}$ from $\mathcal{N}(\mathbf{0}, \mathcal{I})$, and $\{\tilde{z}_1, \tilde{z}_1, ..., \tilde{z}_n\}$ from $Q_\Phi(Z|X)$
    Update weights $\Phi$ and $\Theta$ by descending

$$\frac{1}{n}\sum_{i=1}^n c(x_i, G_\Theta(\tilde{z}_i)) \cdot \lambda \cdot P_Z(\tilde{z}_i) + \frac{1}{n^2 - n}\left(\sum_{h \neq j} k(z_h, z_j) + \sum_{h \neq j} k(\tilde{z}_h, \tilde{z}_j)\right) - \frac{2}{n^2}\sum_{h,j} k(z_h, \tilde{z}_j)$$

**end**
Find $D_{min} = \{Q_\Phi(x_i), Q_\Phi(x_j), ..., Q_\Phi(x_k)\}$, $\|D_{min}\| = (1 - \alpha)\|D\|$ with Minimum Covariance Determinant estimator, $\inf_{\tilde{\Sigma}} Det\{\tilde{\Sigma}\}$.
Find estimated mean $\tilde{\mu}$ from $D_{min}$
**return** $\|Q_\Phi(x_i)\|_{mahalanobis} = (Q_\Phi(x_i) - \tilde{\mu})'\tilde{\Sigma}(Q_\Phi(x_i) - \tilde{\mu})$ for $x_i \in D$ as outlier scores

---

