# OpenReview forum: "Outlier Preserving Distribution Mapping Autoencoders "
_ICLR.cc/2021/Conference — Reject_

### Official Review · AnonReviewer3 · 2020-10-16
**More work is needed to show outliers are mapped to a low probability region.**

**Rating:** 3
**Confidence:** 5

**Review:**

Summary:

The authors proposed a WAE-based algorithm for outlier detection, aiming at mapping outliers to a low probability region and inliers to a high probability region. The training objective is based on WAE by replacing the reconstruction error with the prior weighted loss. Experiments were performed to show the effectiveness of the proposed algorithm.

################

Strong points:

. The authors proposed a training objective, which intends to map outliers to a low probability region.

. The authors theoretically showed that under certain conditions the outliers and inliers can be separated by a distance metric.

################

Weak points:

. In the abstract the authors claimed that ``We formally prove that OP-DMA succeeds to map outliers to low-probability regions''. This is not true. This conclusion is only verified in Experiment 3 using a synthetic dataset. A formal proof in Theorem 2 is regarding the separation between outliers and inliers under a distance metric. As the claim is the main contribution of the paper, more theoretical/experimental work is needed to demonstrate that.

. The authors claimed that the reconstruction error can hardly separate outliers from inliers and showed an example in Figure 1. However, such conclusion may directly depend on the percentage of outliers in the training set. As an extreme case, if the training set only includes inliers, then the reconstruction error may work well in the testing phase.


. In the proposed network architecture, is the encoder deterministic or non-deterministic? It seems that the authors adopt the deterministic one but both Q(X) (deterministic) and Q(Z|X) (non-deterministic) are adopted, which is very confusing.

. The authors claimed that the reconstruction error term in Equation 1 ensures the one-to-one correspondence between input, latent, and reconstructed input. This claim is questionable. Consider the loss function as the L2 norm. It is possible that different reconstructed inputs lead to the same reconstruction error. Also, it is unclear why one-to-one correspondence is critical to outlier detection.

. The expression in Equation (2) is incorrect: the optimization problem should not be subject to P_Q = P_Z as the divergence penalty has already been included.


. The contamination parameter \alpha is required as an input in the algorithm, which significantly limits the practical use of the algorithm. In reality, the value of \alpha can only be estimated, thus inaccurate estimation of \alpha inevitably degrades the algorithm performance. This is also confirmed in Experiment 2: if \alpha is underestimated, F1 score decreases significantly.

. In experiment, the authors compared with benchmarks using F1 score. Another commonly used metric in outlier detection is AUROC. Do the authors have any comparison results regarding that?

################

The paper needs to be proofread. There are many grammar mistakes and typos.

---

### Official Review · AnonReviewer2 · 2020-10-29
**A simple change to autoencoders to preserve outlyingness in latent distribution.**

**Rating:** 4
**Confidence:** 3

**Review:**

Autoencoders suffer from the issue of assigning unusual or outlying samples to regions in the latent space which contain a large amount of nominal samples: i.e. unusual samples are not separated from more normal looking samples. This issue is highly problematic when using an AE for anomaly detection. I this paper the authors present a method to help prevent this using the intuition that samples which are reconstructed poorly are likely to be outlying and incorporating this into the Wasserstein Autoencoder loss. They demonstrate the efficacy of their loss experimentally.

Overall this paper paper suffers from two main problems:

1. Lack of novelty. This slight adjustment to the WAE loss is rather incremental and doesn't incorporate any great new insight or particularly intriguing new concept to the problem, it is a rather small and not-so-interesting adjustment in my opinion.

2. Weak experiments. The authors state
>State-of-the-art deep outlier detection methods fall into one of three categories: 1) Autoencoders coupled with classic outlier detectors (Erfani et al., 2016; Chalapathy et al., 2018), 2) Reconstruction error-based outlier detection methods (Zhou & Paffenroth, 2017; Chen et al., 2017; Sabokrou et al., 2018; Xia et al., 2015), or 3) Generative outlier detection methods (Perera et al., 2019; Vu et al., 2019; Liu et al., 2019).

Which is a woefully dated outlook of deep outlier detection, considering predominant  SOTA methods include self-supervised learning (Deep Anomaly Detection Using Geometric Transformations Golan and Yaniv) and deep one class models (Ruff et al. 2018), and more rich probabliistic models (DEEP ANOMALY DETECTION WITH OUTLIER EXPOSURE Hendrycks et al 2019). Representatives from these classes of outlier detection models should really be included. Additionally the authors experiments are predominantly on "non-deep" (i.e. low dim non-image) datasets which are of limited interest for deep models. This is further exacerbated by the fact that standard blackbox "classic" outlier detection methods are missing (e.g. iForest) and details are missing about other classic methods, e.g. what is the kernel for the OC-SVM.

Due to the above considerations I cannot recommend that this paper is accepted to a top tier ML conference in its current state.

---

### Official Review · AnonReviewer4 · 2020-10-29
**The paper focuses on the problem of outlier detection based on the decomposition of original data into inliers and outliers.**

**Rating:** 5
**Confidence:** 3

**Review:**

The paper proposes a solution for outlier detection implicitly based on the idea of robust principal component analysis that decomposes data into inliers and outliers while utilizing a novel prior weighted loss and addressing two challenges: overpowering divergence penalty and unknown outlier status.

##Summary of the main contributions:
(1) A novel Autoencoders that distinguishes inliers and outliers by optimizing Wasserstein distance
(2) A novel probability weighting approach to address two aforementioned challenges

##Overall Feedback
I found the paper easy to follow, which means it is well structured. However, the level of novelty is not high. In other words, the proposed solution combines several existing approaches to tackle the problem and address the challenges. The experimental settings (datasets, experiments and metric) is strong, but more recent baselines can be included, especially those based on auto encoders. So, the key concern is lack of through experimental evaluation to study the effectiveness of the proposed method.

##Strong Points
(1) The proposed method is technically correct


##Weak Points
(1) The level of novelty is not high
(2) The proposed method does not work in end-to-end manner. More precisely, the main focus is on the input in an appropriate format for traditional outlier detection method. However, it is often well-observed that joint methods work better in reality.
(3) Experimental evaluation requires improvement. For instance, more details and elaboration is necessary for Experiment 3 which seems to be important for supporting the claim that the paper makes.

##Suggestions:
(1) Please elaborate more on the proposed loss function, especially the Autoencoders loss
(2) Please include more baselines on outlier detection into the experimental evaluation

##Minor comments:
(1) resolution of figure is not high

---

### Official Review · AnonReviewer1 · 2020-11-03

**Rating:** 6
**Confidence:** 4

**Review:**

The paper proposes an autoencoder-based outlier detection system. The main idea of the paper is to ensure that outlier points are mapped to areas distant from the inliers in the embedding space. To this end, a novel cost function is introduced, which weighs the reconstruction error based on a prior distribution for the embedding space. This cost function hopes to force inliers to be mapped to the high probability region of the prior distribution and push outliers to low probability regions. Combined with a normal/multivariate prior distribution, this then enables the use of simple distance-based outlier detection methods.

While the idea and goal of the paper are sensible specific claims and assumptions are not explained or motivated well. For example, the paper claims that at the beginning of the training, the reconstruction error of outliers will be higher than that of inliers. It is unclear why, for an untrained network, this should be true. This is a core assumption of the paper and yet no detail about this aspect it is given. The fact that a distribution loss over batches is employed might aid in achieving the goal, however, a more substantial discussion and theoretical underpinning of this core claim are needed.

The complete method is never described in detail. The closest is Section 3.4, where all the components are quickly enumerated. It would have been desirable to see a discussion of the choices made and what properties they provide and possibly what alternatives could be used.

The theoretical results provided are somewhat confusing. For one, the main paper contains only the theorems and none of the proofs or sketches thereof. This means that the paper is only complete with its appendix. Theorem 2 indicates that the optimal solution for the algorithm is to map outliers to low probability areas. However, this does not guarantee that the embedding being learned will actually do so. It would have been an excellent addition to talk about this and if there are any mechanisms that could be used to check if the learned embedding is indeed performing as desired.

The experiments demonstrate that the proposed method works very well and better than other methods. However, sadly the paper lacks any meaningful discussion of the results or insights to be gained from them. Good results are encouraging, however, they do not make up for a lack of discussion. For example, the few cases were the proposed method is not performing on par with other methods it would be good to know why that is. Another interesting aspect is what is so special about the Lympho dataset that a handful of methods perform exceptionally well while the others fare much worse. The description of experiment-2 is very unclear, and it is not clear at all what the experiment tries to showcase or what the outcome was.

While the datasets chosen provide a good variability in scenarios, an interesting addition would have been synthetic data which explores various aspects, such as types of outliers, types of inlier distributions, etc. Such a study of the properties or behaviour of the proposed method in controlled environments would be a great addition to understand the method in greater detail.

---

### Decision · Program_Chairs · 2021-01-07
**Final Decision**

**Decision:**

Reject

**Comment:**

In this paper, a data mapping method to a latent space designed for outlier detection is proposed. Outlier detection by latent space mapping has been extensively studied in the literature. Unfortunately, this paper does not fully discuss the relation of the proposed method with a large amount of existing literature and lacks novelty.